# Five-Year Follow-Up of a Child with Non-Syndromic Oligodontia from before the Primary Dentition Stage: A Case Report

**DOI:** 10.3390/children10040717

**Published:** 2023-04-12

**Authors:** Tatsuya Akitomo, Satoru Kusaka, Yuko Iwamoto, Momoko Usuda, Mariko Kametani, Yuria Asao, Masashi Nakano, Meiko Tachikake, Chieko Mitsuhata, Ryota Nomura

**Affiliations:** 1Department of Pediatric Dentistry, Graduate School of Biomedical & Health Sciences, Hiroshima University, Hiroshima 734-8553, Japan; 2Department of Pediatric Dentistry, Hiroshima University Hospital, Hiroshima 734-8553, Japan

**Keywords:** non-syndromic oligodontia, congenital absence of teeth, dental arch, dimensional changes

## Abstract

Background: Congenital absence of teeth is a major dental abnormality in pediatric dentistry and the absence of six or more teeth is defined as oligodontia. Few reports of patients with non-syndromic oligodontia without systemic disease have continued dental follow-up from an early age. Methods: We performed the five-year follow-up from before the eruption of the primary dentition of a Japanese child with non-syndromic oligodontia and analyzed changes in dental arch growth. Results: At the oral examination at the age of 1 year and 2 months, eight primary incisors were congenitally absent. Therefore, we made dentures for the patient at the age of 3 years and 4 months. From the age of 5 years and 1 month, the child received articulation training for dysarthria from a speech therapist to improve the function and appearance of the oral cavity. Measurement of the patient’s dental models revealed a particularly narrow dental arch, especially between the primary canines. Conclusions: Our findings highlight the importance of treatment for patients with non-syndromic oligodontia from an early age by multiple medical professionals, recognizing that the missing teeth affect the growth of the maxillofacial region.

## 1. Introduction

Abnormalities in the number of teeth can be caused by a variety of factors, including heredity, problems during tooth germ formation, drugs, and maternal smoking during pregnancy [1,2,3,4,5]. Congenital absence of teeth is one of the most common dental abnormalities in tooth development, with a frequency ranging from 2.6% to 11.3% [6,7,8]. In contrast, the congenital absence of teeth in the primary dentition is rare, with a reported incidence of 0.5% to 0.9% [9,10,11].

The congenital absence of teeth is classified into hypodontia and oligodontia. Hypodontia is defined by agenesis of fewer than six teeth, while oligodontia is a congenital absence of six or more permanent teeth, excluding third molars [12]. Some cases with oligodontia are associated with ectodermal dysplasia, cleft lip, cleft palate, and other syndromes, but non-syndromic cases also occur [13,14].

Infants and children may experience problems that affect nutrition, such as oligodontia and ankyloglossia [15,16]. These abnormalities often have a significant impact on development and require appropriate treatment by dentists. Patients with oligodontia often use dentures to restore mastication, pronunciation, and esthetics [17], while patients with ankyloglossia can undergo frenotomy to improve tongue function [16].

Oral management of children with non-syndromic oligodontia from the time when no primary teeth have erupted has rarely been reported. Additionally, there are no reports about dental arch morphology and dimensional changes with growth in the primary dentition in children with non-syndromic oligodontia. We monitored the follow-up of a child with non-syndromic oligodontia, and measured the width of the dental arch each time a denture was made. Here, we report the oral management over a period of 5 years of a child with non-syndromic oligodontia in the primary dentition and the effect of the missing teeth on the maxillofacial region.

## 2. Case Presentation

A 14-month-old Japanese boy presented with delayed tooth eruption to the Department of Pediatric Dentistry of Hiroshima University Hospital. His mother and maternal relatives had family histories of congenital absence of primary teeth, while his father and siblings had no such symptoms. Intraoral examination revealed that there were no primary teeth in the oral cavity (Figure 1A). Dental X-ray images showed congenital absence of the bilateral primary central and lateral incisors of the maxilla and mandible (Figure 1B). However, all primary canines were found in the X-ray images. We instructed the patient’s guardian to modify the patient’s eating habits to prevent choking because of the absence of the primary anterior teeth, and continued to monitor the eruption of the other primary teeth. Follow-up appointments established that the first primary molar began to erupt at 1 year and 7 months, the primary canine at 2 years and 1 month, and the second primary molar at 2 years and 6 months (Figure 2).

Although the patient had no abnormality in his hair or skin, the possibility of ectodermal dysplasia was investigated because of the congenital absence of multiple teeth. The patient was referred to the Department of Pediatrics of Hiroshima University Hospital at 1 year and 7 months. Height and weight were both slightly low, but there was no evidence of systemic disease. Bilateral permanent central and lateral incisors of the maxilla and mandible were not detectable in the dental X-ray images at 2 years and 10 months and 3 years and 8 months, indicating that they were congenitally absent (Figure 3). As a result, the patient was diagnosed with non-syndromic oligodontia without systemic disease.

As the patient grew up, the esthetic and occlusal interference became more pronounced, as did the dysarthria of the [s] speech sound. Because he was cooperative with dental treatment, we made dentures at the age of 3 years and 4 months. He has used his denture at all times except when sleeping and toothbrushing (Figure 4). Additionally, he received articulation training from a speech therapist from the age of 5 years and 1 month, resulting in improvement of his dysarthria.

At the age of 6 years and 3 months, the mandibular right first molar began to erupt (Figure 5A,B). We have provided ongoing oral hygiene instruction and dietary advice since the age of 1 year and there have been no findings of dental caries or gingivitis. The panoramic radiograph shows the congenital absence of all the permanent teeth where the primary teeth were congenitally absent, as well as the absence of the tooth germs of the maxillary left canine and second premolar (Figure 5C).

Dental models of the maxillary and mandibular dental arches were made at the ages of 3 years and 4 months, 4 years and 7 months, and 5 years and 9 months for fabrication of dentures. We compared the tooth width and dental arch width with data from a typical Japanese male [18] (Table 1). The maxillary second primary molars were 1 SD smaller than those of the mean Japanese male.

The dental arch measurements are shown in Table 2 and the chronological changes in the values are summarized in Table 3 and Table 4. The dental arch widths for 3 years and 4 months are smaller than those of typical Japanese males aged 3–4 years for all items, and both the C_C_–C_C_ and C_L_–C_L_ of the maxilla and mandible were 3 SDs smaller. Although the patient’s arch measurements were increasing with growth, even at 5 years and 9 months many items were smaller than those for typical Japanese males aged 3–4 years. At 5 years and 9 months, both the C_C_–C_C_ and C_L_–C_L_ of the maxilla were 2 SDs smaller compared with those of typical Japanese males aged 5 years. Furthermore, the C_C_–C_C_ and C_L_–C_L_ of the mandible were 1 SD and 2 SDs smaller, respectively, and the D–D of the maxilla was 1 SD smaller.

## 3. Discussion

Non-syndromic oligodontia is rare, with a reported prevalence of approximately 0.08% to 0.16% [19,20]. Oligodontia requires rehabilitation of the dental arches through orthodontic, prosthetic, and implant treatment to meet the patient’s esthetic and stomatognathic functional needs [21]. In the present case, a 1-year-old child with non-syndromic oligodontia was followed continually from before the eruption of teeth and throughout the primary dentition period.

At the age of 3 years and 4 months, the patient had a large defect in the anterior dental region, which had caused dysarthria of the [s] speech sound. Kalia et al. (2018) reported that prosthetic rehabilitation with a fixed functional space maintainer in children with missing maxillary anterior teeth improves pronunciation of the [v], [ph], [d], [dh], [th], [t], [s.], and [s] speech sounds [22]. Therefore, we decided to make dentures and commence articulation training, including tongue movement, with a speech pathologist. The patient initially resisted the use of dentures, but he was highly motivated by the improved esthetics of the dentures and subsequently became a long-term user. The articulation training from the age of 5 years and 1 month resulted in early improvement of his dysarthria. Gonçalves et al. (2013) reported that multidisciplinary treatment, including speech and prosthetic therapy, restores masticatory function and esthetics, allowing the patient to achieve greater self-esteem and better social acceptance [23]. Thus, we highlight the importance of multidisciplinary collaboration for the treatment of patients with non-syndromic oligodontia.

Children without anterior teeth are at risk of developing abnormal tongue habits [24]. Morphological or functional abnormalities of the tongue can cause problems with breastfeeding and nutritional intake [16]. Such anomalies can also lead to mental disorders and may affect future oral development and function [24]. In our case, tongue training and denture fabrication at an early age may have prevented tongue dysfunction and nutritional intake problems.

Dental model measurements revealed no obvious differences between the patient and the control in the width of each tooth except for the second primary molar, indicating that non-syndromic oligodontia has little effect on the development of the teeth that are present. However, the dental arch at 3 years and 4 months was narrow, especially in the space between the primary canines. The congenital absence of primary teeth reduces the arch length available for eruption of the permanent teeth, resulting in a predisposition for crowding, rotation, and impaction of the permanent teeth [25]. The absence of anterior teeth may also cause the canines to erupt with a mesial inclination, creating a narrow arch.

Although the width of the dental arch of the patient increased with growth, the increase was smaller than that of the typical Japanese male. Even at the age of 5 years and 9 months, all items except D–D and E–E of the mandible were smaller than average for a 3–4-year-old male. Our results suggest that the congenital absence of multiple incisors may be closely related to growth inhibition of the dental arch. Hsu et al. (1998) reported that skeletally narrowed maxillary posterior widths and dentally widened mandibular posterior widths were found in patients with anterior open bite [26]. The present case resembled the condition of an anterior open bite and had similar morphological characteristics of the dental arches (narrow maxilla and wide mandible), indicating that congenital absence of incisors causes the same kind of malocclusion as anterior open bite and changes the dental arch morphology in the primary dentition.

Mochizuki et al. (1965) reported that the width between the primary canines increases significantly in the permanent incisor eruption period to make space for the permanent incisors [27]. In contrast, it was reported that patients with bilateral maxillary lateral incisor agenesis have a significantly reduced ANB angle and maxillary length [28]. In the present case, although eight permanent incisors were congenitally missing, the use of dentures was expected to prevent the space from decreasing further. However, the lateral growth would be expected to weaken in the permanent incisor eruption period.

In cases with congenitally missing lateral incisors, early intervention to enhance the growth of the maxilla may avoid the development of a skeletal class III malocclusion and may prevent the need for more invasive interventions later in life [28]. Additionally, in patients with anterior open bite, lateral expansion of the maxilla is recommended as orthodontic treatment [26]. This expansion of maxilla may also be recommended for patients with bilateral congenitally missing lateral incisors with anterior open bite. However, orthodontic treatment such as rapid expansion will increase the space for oligodontia. Therefore, it is important that pediatric dentists, orthodontists, and prosthodontists work together to achieve optimal long-term treatment outcomes.

We followed this patient with non-syndromic oligodontia from before the normal eruption time of the primary teeth and measured changes in dental arch growth. Non-syndromic oligodontia is rare and we focused on only one child in this case report. Because few patients present with non-syndromic oligodontia, it would be desirable to conduct large studies in cooperation with other hospitals in the future. Additionally, this study only followed the patient from before the eruption of the primary teeth through the primary dentition stage; future case studies should extend through to the late mixed dentition stage.

## 4. Conclusions

We conducted a five-year follow-up of a patient with non-syndromic oligodontia starting at an early age, before the patient’s teeth had erupted. This case highlights the importance of the involvement of multiple medical professionals for patients with oligodontia, including denture maintenance and articulation training by a speech therapist. Additionally, dental model measurements confirmed the narrow width of the dental arch in non-syndromic oligodontia patients, suggesting that the absence of multiple incisors affects dental arch growth.

## Figures and Tables

**Figure 1 children-10-00717-f001:**
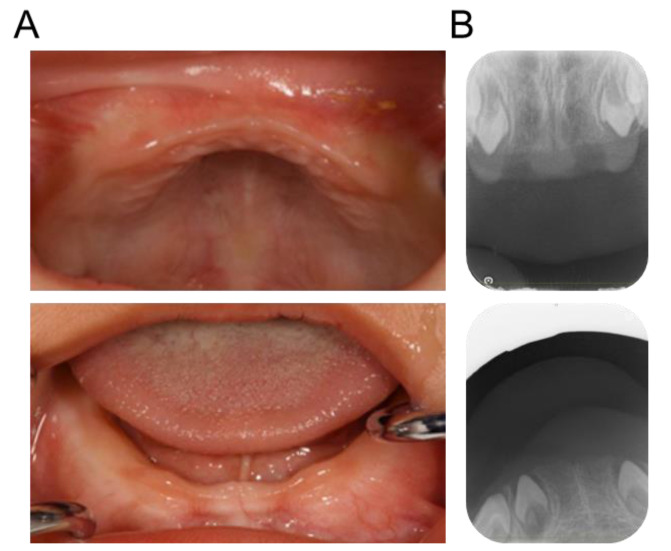
Intraoral findings at first visit (1 year and 2 months). (**A**) Intraoral photographs. (**B**) Periapical radiographs.

**Figure 2 children-10-00717-f002:**
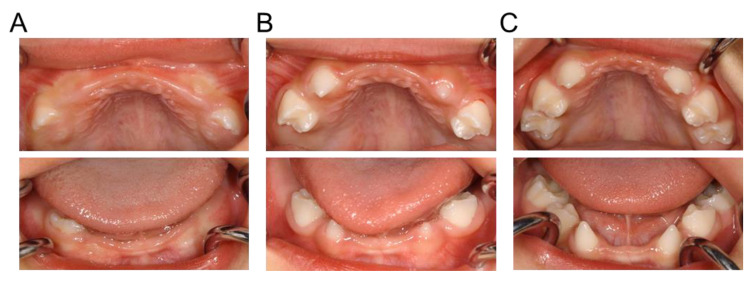
Intraoral photographs during tooth eruption. (**A**) First primary molar eruption (1 year and 7 months). (**B**) Primary canine eruption (2 years and 1 month). (**C**) Second primary molar eruption (2 years and 6 months).

**Figure 3 children-10-00717-f003:**
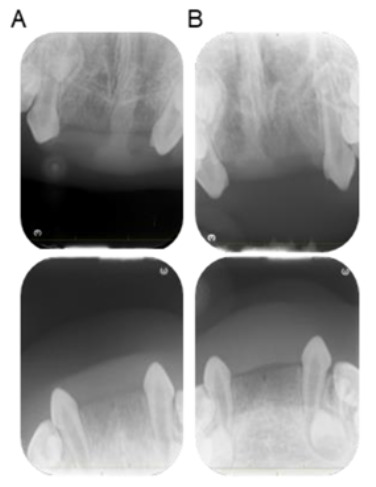
Periapical radiographs of the incisor region. (**A**) 2 years and 10 months. (**B**) 3 years and 8 months.

**Figure 4 children-10-00717-f004:**
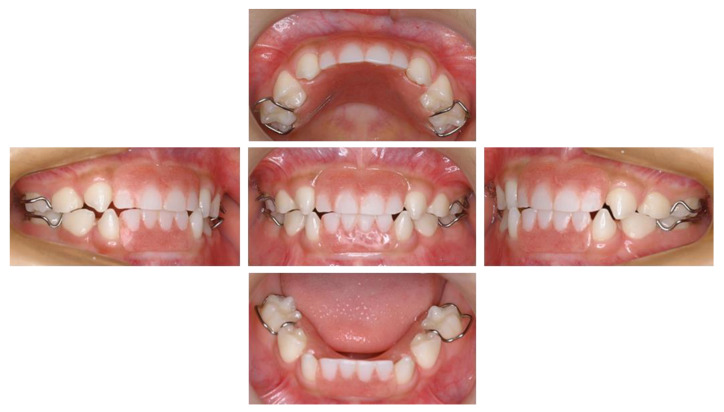
Intraoral photographs with denture at the age of 4 years and 11 months.

**Figure 5 children-10-00717-f005:**
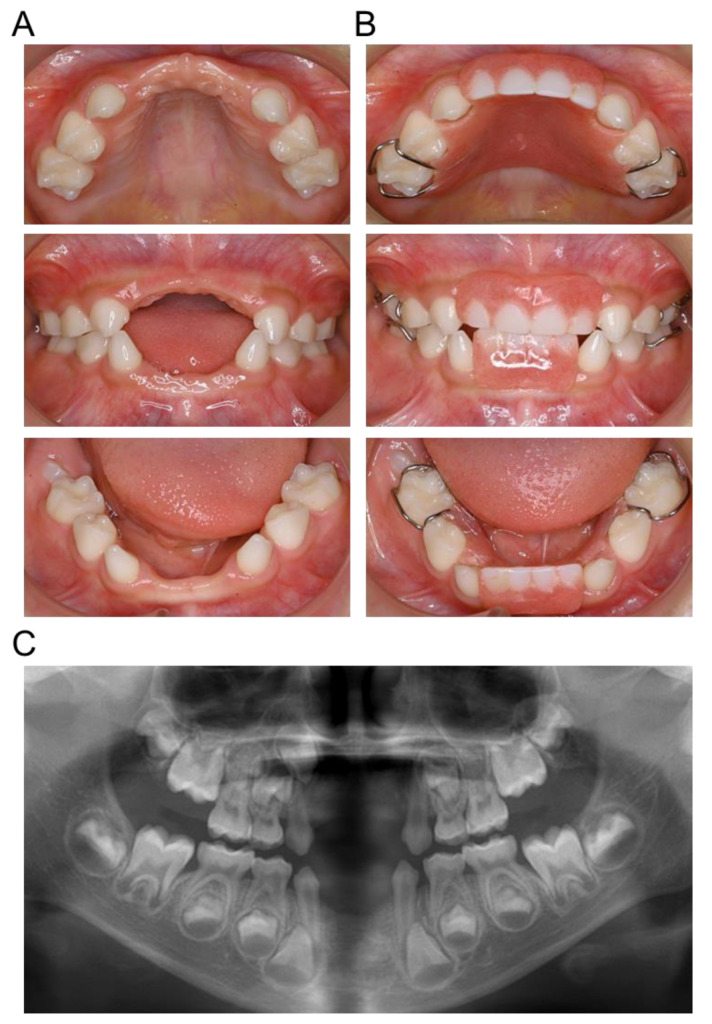
Intraoral findings at 6 years and 3 months. (**A**) Intraoral photographs without denture. (**B**) Intraoral photographs with denture. (**C**) Panoramic radiographs.

**Table 1 children-10-00717-t001:** Width of primary teeth.

**Maxilla**		
Tooth	Present case (mm)	Mean ± 1 SD [18] (mm)
C	6.78	6.67 ± 0.43
D	7.38	7.36 ± 0.41
E	8.58 *	9.30 ± 0.41
**Mandible**		
Tooth	Present case (mm)	Mean ± 1 SD [18] (mm)
C	5.85	5.82 ± 0.34
D	8.08	8.23 ± 0.48
E	10.03	10.34 ± 0.51

* Lower than 1 SD compared with the mean.

**Table 2 children-10-00717-t002:** Dental arch measurement items.

**Maxilla**	
Item	Distance
C_C_–C_C_	Distance between cusps of maxillary bilateral primary canines
C_L_–C_L_	Distance between lowest point of palatal cervical area of maxillary bilateral primary canines
D–D	Distance between buccal cusps of maxillary bilateral first primary molars
E–E	Distance between buccal terminal sulci of maxillary bilateral second primary molars
**Mandible**	
Item	Distance
C_C_–C_C_	Distance between cusps of mandibular bilateral primary canines
C_L_–C_L_	Distance between lowest point of lingual cervical area of mandibular bilateral primary canines
D–D	Distance between buccal terminal sulci of mandibular bilateral first primary molars
E–E	Distance between mesial buccal sulci of mandibular bilateral second primary molars

**Table 3 children-10-00717-t003:** Dimensional changes of the dental arch at the age of 3–4 years.

**Maxilla**			
Item	3y4m (mm)	4y7m (mm)	Average of 3–4 years old [18] (Mean ± 1 SD, mm)
C_C_–C_C_	23.4 ***	24.9 ***	30.4 ± 1.5
C_L_–C_L_	19.2 ***	20.6 ***	24.8 ± 1.3
D–D	36.1 *	36.9 *	39.5 ± 2.0
E–E	44.1 *	45.0	46.6 ± 2.0
**Mandible**			
Item	3y4m (mm)	4y7m (mm)	Average of 3–4 years old [18] (Mean ± 1 SD, mm)
C_C_ –C_C_	19.0 ***	20.8 *	23.4 ± 1.3
C_L_–C_L_	14.7 ***	16.0 **	19.1 ± 1.2
D–D	32.4	32.8	33.4 ± 1.5
E–E	37.7	38.6	39.0 ± 1.8

Lower than * 1 SD, ** 2 SD and *** 3 SD compared with the average.

**Table 4 children-10-00717-t004:** Dimensional changes of the dental arch at the age of 5 years.

**Maxilla**		
Item	5y9m (mm)	Average of 5 years old [18] (Mean ± 1 SD, mm)
C_C_–C_C_	25.6 **	31.2 ± 1.9
C_L_–C_L_	21.2 **	25.5 ± 1.8
D–D	37.2 *	40.3 ± 2.7
E–E	45.1	47.6 ± 2.8
**Mandible**		
Item	5y9m (mm)	Average of 5 years old [18] (Mean ± 1 SD, mm)
C_C_–C_C_	21.6 *	23.9 ± 1.7
C_L_–C_L_	16.7 **	19.8 ± 1.3
D–D	33.4	34.3 ± 2.2
E–E	39.4	39.6 ± 2.4

Lower than * 1 SD and ** 2 SD compared with the average.

## Data Availability

Not applicable.

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
