# Peer review of "Five-Year Follow-Up of a Child with Non-Syndromic Oligodontia from before the Primary Dentition Stage: A Case Report"

_children, 2023, doi:10.3390/children10040717_

Round 1

Reviewer 1 Report

This paper reports a case report of a child with non-syndromic oligodontia (NSO).

The authors claimed that the long-term follow-up is rare for NSO.

Actually this is not the case and the long-term is not long enough.

And the case is only one. Therefore, the measurement cannot be generalized at all.

Please include more patients and follow-up long enough to include late mixed dentition at least.

Author Response

The authors claimed that the long-term follow-up is rare for NSO. Actually this is not the case and the long-term is not long enough. And the case is only one. Therefore, the measurement cannot be generalized at all. Please include more patients and follow-up long enough to include late mixed dentition at least.

(Response)

We agree with the reviewer’s comments. We have modified the “long term” to “five-year” in the title of the revised manuscript. We have also removed the words “long term” throughout the revised manuscript. Focusing on only one case of a child with non-syndromic oligodontia and not following up through to the late mixed dentition stage is a limitation. We have discussed the limitations in lines 196-203 of the Discussion section of the revised manuscript as follows. “We followed this patient with non-syndromic oligodontia from before the normal eruption time of the primary teeth and measured changes in dental arch growth. Non-syndromic oligodontia is rare, and we focused on only one child in this case report. Because few patients present with non-syndromic oligodontia, it would be desirable to conduct large studies in cooperation with other hospitals in the future. Additionally, this study only followed the patient from before the eruption of the primary teeth through the primary dentition stage; future case studies should extend through to the late mixed dentition stage.”

Reviewer 2 Report

Dear Authors, this paper about Long-term follow-up of a child with non-syndromic oligodontia from before the primary dentition stage is really interesting and well performed. I am pretty sure that both scientists and dental professionals will find it helpful. By the way, some issues need to be solved before its final acceptance in the paper.

Abstract: please divide it into: introduction, materials and methods, results, conclusions.

Introduction: introduction is a really important part of a scientific article, it helps the reader to deep into the subject you are presenting. In your article, this part is too short and it needs to be lengthened. Please add a small chapter about how oligodontia and ankyloglossia can affect nutrition in infants and children. This paper can help you: Colombari GC, Mariusso MR, Ercolin LT, Mazzoleni S, Stellini E, Ludovichetti FS. Relationship between Breastfeeding Difficulties, Ankyloglossia, and Frenotomy: A Literature Review. J Contemp Dent Pract. 2021 Apr 1;22(4):452-461.

Case presentation: if you want to maintain the “case presentation” phrase, you should add in the title: a case report. If not, please substitute “case presentation” with “materials and methods”

Discussion: this parts is fine and well discussed, I would suggest to discuss some more what has been added in the introduction: the relationship existing between tongue abnormalities and infant nutrition

Author Response

Dear Authors, this paper about Long-term follow-up of a child with non-syndromic oligodontia from before the primary dentition stage is really interesting and well performed. I am pretty sure that both scientists and dental professionals will find it helpful. By the way, some issues need to be solved before its final acceptance in the paper.

Abstract: please divide it into: introduction, materials and methods, results, conclusions.

(Response)

In accordance with the reviewer’s opinion and other case reports in Children, we have added “Background”, “Methods”, “Results”, and “Conclusions” to the Abstract of the revised manuscript.

Introduction: introduction is a really important part of a scientific article, it helps the reader to deep into the subject you are presenting. In your article, this part is too short and it needs to be lengthened. Please add a small chapter about how oligodontia and ankyloglossia can affect nutrition in infants and children. This paper can help you: Colombari GC, Mariusso MR, Ercolin LT, Mazzoleni S, Stellini E, Ludovichetti FS. Relationship between Breastfeeding Difficulties, Ankyloglossia, and Frenotomy: A Literature Review. J Contemp Dent Pract. 2021 Apr 1;22(4):452-461.

(Response)

We have added the following sentences to lines 38-42 of the Introduction of the revised manuscript: “Infants and children may experience problems that affect nutrition, such as oligodontia and ankyloglossia [Nirmala et al., Int J Clin Pediatr Dent, 2013; Colombari et al., J Contemp Dent Pract, 2021]. These abnormalities often have a significant impact on development and require appropriate treatment by dentists. Patients with oligodontia often use dentures to restore mastication, pronunciation, and esthetics [He et al., J Esthet Restor Dent, 2007], while patients with ankyloglossia can undergo frenotomy to improve tongue function [Colombari et al., J Contemp Dent Pract, 2021].”

Case presentation: if you want to maintain the “case presentation” phrase, you should add in the title: a case report. If not, please substitute “case presentation” with “materials and methods”

(Response)

We have added “a case report” to the title of the revised manuscript.

Discussion: this parts is fine and well discussed, I would suggest to discuss some more what has been added in the introduction: the relationship existing between tongue abnormalities and infant nutrition

(Response)

We have added the following sentences to lines 155-160 of the Discussion section of the revised manuscript. “Children without anterior teeth are at risk of developing abnormal tongue habits [Mohammad et al., Int J Clin Pediatr Dent, 2018]. Morphological or functional abnormalities of the tongue can cause problems with breastfeeding and nutritional intake [Colombari et al., J Contemp Dent Pract, 2021]. Such anomalies can also lead to mental disorders and may affect future oral development and function [Mohammad et al., Int J Clin Pediatr Dent, 2018]. In our case, tongue training and denture fabrication at an early age may have prevented tongue dysfunction and nutritional intake problems.”

Round 2

Reviewer 1 Report

The manuscript is improved within the limitations.